# Simultaneous Generation and Improvement: A Unified RL Paradigm for FJSP Optimization

## Abstract

We present an end-to-end reinforcement learning framework designed to address the Flexible Job Shop Problem (FJSP). Our approach consists of two primary components: a generative model that produces problem solutions stepwise, and a secondary model that continually refines these (partial) solutions. Importantly, we train both models concurrently, enabling each to be cognizant of the other's policy and make informed decisions. Extensive experimentation demonstrates that our model delivers better performance in shorter time on several public datasets comparing to baseline algorithms. Furthermore, we highlight the superior generalizability of our approach, as it maintains strong performance on large-scale instances even when trained on small-scale instances. It is worth noting that this training paradigm can be readily adapted to other combinatorial optimization problems, such as the traveling salesman problem and beyond.

## 1 Introduction

In the field of industrial manufacturing, the scheduling of manufactured demands and manufacturing resources is an extremely important task. The task requires the scheduling system to provide an optimal scheduling solution as much as possible so that all manufactured demands can be completed in the shortest time on the processing machine. Cloud manufacturing that has developed in recent years, due to the open environment of the cloud foundation, these scheduling problems will become larger in scale, more diversified and more dynamic. These changes have put higher demands on the real-time and versatility of the scheduling system.

This paper mainly studies a well-known NP-hard problem —- Flexible Job Shop Scheduling Problem which is famous in the field of cloud manufacturing. FJSP is an extension of a well-known combinatorial optimization problem which is Job Shop Schdule Problem(JSSP). Unlike the operations in the JSP problem that can be processed on any machine, the operations in FJSP are only allowed to be processed on their respective machine sets. This peculiarity makes FJSP not only need to consider the processing order of operations, but also need to consider assigning suitable processing machines for operations. These factors make FJSP more complex and diversified.

Due to the complexity of the FJSP problem, exact solution methods such as mathematical programming and constraint programming require a lot of computational time. The computational time will become unbearable when the instance size is too large. Therefore, some approximate solution methods that reduce solution quality in exchange for solution time have been proposed.

Priority dispatching rule(PDR) is a well-known heuristic method [4] that schedules operations to suitable machines for processing by designing a series of priority rules, such as algorithms based on FIFO(first in first out) and SPT(shortest processing time).

Meta-heuristic methods represented by genetic algorithms [7; 8; 9; 10] have also received much attention. Compared with PRD heuristic methods, these methods can get higher quality solutions but also require more time.

With the increasing attention paid to deep learning and reinforcement learning in the field of combinatorial optimization in recent years, some related methods [28; 29] have also been used to solve FJSP problems. The method based on deep reinforcement learning will establish an MDP model for the FJSP problem and then train a parameterized policy model in this MDP model through a

human-designed reward function. Finally, use the trained policy model to solve the FJSP problem. Compared with traditional methods, these reinforcement learning-based methods have shown great potential in terms of computational time and solution quality.

In this paper, we propose a new end-to-end framework based on graph neural networks and deep reinforcement learning for solving FJSP problems. Our framework consists of two models: one is a generation model used to generate scheduling solutions for unassigned operations; the other is an improving model used to improve current assigned operations scheduling solutions. When solving, these two models will make decisions alternately. The generation model generates a partial solution, and then the improvement model improves this partial solution to output an improved partial solution. Then the generation model will continue to generate more complete partial solutions based on the improved solution of the previous round until all workpieces are completed assigned, and finally the optimal solution output by the improvement model for the last time will be used as the final solution.

Due to our framework using graph neural networks to extract features from FJSP instance, there is no limit to the scale size of input instances for our model. This peculiarity allows our model to solve instances of different scales rather than just training set scales. This allows us to train our model on small-scale instances data set and then use it to solve larger-scale problem. This powerful generalization ability allows our model to have a wider range of application scenarios.

We conducted comparative experiments on public datasets and randomly generated datasets to verify that our proposed method has more advantages in terms of solving time and efficiency compared with other methods. Not only that, we also conducted a series of ablation experiments to study the impact of each module in our proposed framework on experimental results. In summary, this article makes the following main contributions:

- Proposed an end-to-end framework based on graph neural networks and deep reinforcement learning for solving FJSP problems.
- Designed a novel joint training method for training our proposed framework.
- Conducted a series of ablation experiments to study each module in our proposed framework.

## 2   BACKGROUND

**Flexible Job Shop Schedule Problem**   A FJSP instance of size $n \times m$ includes $n$ jobs and $m$ machine. Using $\mathcal{J}$ to represent the set of job and $\mathcal{M}$ to represent the set of machine. For each job $J_i \in \mathcal{J}$ has an operation set $\mathcal{O}_i$,which contains $n_i$ operations,$O_{ij}$ means the operation with processing order $j$ of the $J_i$ in $\mathcal{J}$. Each operation $O_{ij}$ can be processed on any machine $M_k$ from its compatible set $\mathcal{M}_{ij} \in \mathcal{M}$ for a processing time $p_{ijk}$ when the machine is in idle state.To obtain a solution for FJSP,each operation $O_{ij}$ should be assigned a machine and start processing time $S_{ij}$ to process. A FJSP instance can be described as a disjunctive graph $\mathcal{G} = (\mathcal{O}, \mathcal{C}, \mathcal{D})$. Specifically $\mathcal{O} = \{O_{ij} | \forall i, j\} \cup \{start, end\}$ is the node set,which includes all operation and two dummy nodes means start and end of production.These two node can be processed by any machine with zero processing time. $\mathcal{C}$ is the set of conjunctive arcs,which are directed arcs that form $n$ paths from $Start$ to $End$ representing the repective processing sequence of $J_i$. $\mathcal{D} = \cup_k \mathcal{D}_k$ is the set of disjunctive arcs which are undirected,and $\mathcal{D}_k$ is a set of arcs which connects the operations that can be processed on machine $M_k$. Solving FJSP is equivalent to selecting a disjunction arc and fixing its direction for each operation node. In order to more clearly represent the belonging relationship of each operation to the machine in the solution, we separate these directional displacement arcs from the disciple graph to form a solution graph, which will be used to serve our framework for solution generation and improvement. A FJSP instance and one of its feasible solution can be described by Fig 1.

**Conventional Method for FJSP**   Coneventional methods for solving the FJSP problem can be roughly divided into three categories: exact methods, heuristic methods, and meta-heuristic methods.Exact methods usually combine integer linear programming [3] and constraint programming [4]. These methods can guarantee the optimality of the solution through rigorous mathematical reasoning, but they require a lot of computational time. Therefore, exact methods can only be used to solve small-scale problems. Heuristic methods design optimization algorithms by introducing expert prior knowledge. Compared to exact methods, heuristic methods require very short computational

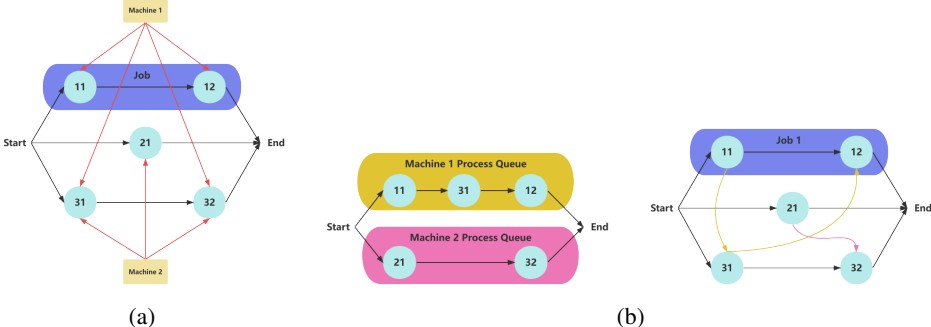

(a)                                    (b)

Figure 1: (a) An FJSP instance of size $3 \times 2$. (b)Two different representations of the same feasible solution.

time, but at the same time, they also reduce the quality of the solution. Typical heuristic methods for solving the FJSP problem include PDR [5], A* [6], local search [7], etc. More Specially, the methods based on PDR are widely used in production systems in real world due to its easy implementation and less computation time. Meta-heuristics can be further classified into single solution based(e.g. simulated annealing [8], tabu search [9] and population-based (e.g. genetic algorithm [10; 11]) methods, which works on a single solution or a population of solutions. Complete reviews of FJSP methods can be found in [1; 2].

**DRL base method for FJSP**     Recently,more and more work is using DRL to solve complex scheduling problems. The key issues when using DRL are how to extract features from the FJSP and how to formulate the problem as an MDP. Some research use multilayer perceptron (MLP) [12; 13; 14; 15] or convolution neural network (CNN) [16] to extract features from FJSP instance to represent the state as a feature vector. It is obvious that these methods have a significant disadvantage that the network structure will limit the size of the input problem, which means that a trained model can only solve FJSP instances of the same size. Some methods based on recurrent neural network(RNN) [32]and attention mechanism [18] are proposed to solve the problem of different input sizes. Obviously, compared with the above-mentioned network structures, GNN [20] has a natural advantage in dealing with FJSP problems represented by graphs. [21] propose methods which combine DRL and GNN to learn PDRs for JSP. [23] uses DQN to solve FJSP by selecting the best one from a pool of hand-crafted PDRs with policy model. [24] propose an DRL method without considering the graph structure which makes it lose a lot of useful information for decision making. [29; 30] propose an end-to-end DRL method combines GNN respectively,which generate solutions of FJSP by assigning operations step-by-step using a policy model.We will compare our method with the methods of these two papers in the experimental section.

## 3 GENERALIZATION-IMPROVING MODEL

We use a Graph $\mathcal{S} = (\mathcal{O}, \mathcal{E}, \mathcal{C})$ to represent the solution of FJSP (in figure). It should be noted each arc $E \in \mathcal{E}$ connects two sequentially adjacent operations that are processed on the same machine and Each arc $C \in \mathcal{C}$ connects the operation on the same job,which both of these two arc set are directed. There are at most $m$ forward paths from start to end in $\mathcal{S}$ ($m$ is the number of machines) which are connected by arc in $\mathcal{E}$ , and each path represents the operation processing queue of a machine. More specifily, if $O_{ij}$ is located on path $k$ with position $l$, it means $O_{ij}$ has been assigned to $M_k$ with order $l$ to process. We use makespan $C_{max}$ to measure the quality of a solution $\mathcal{S}$, which is equal to the processing completion time of the last operation to be completed among all operations.

To solve FJSP, we need to generate $\mathcal{S}$ that makes all operation connect with directed arcs $E \in \mathcal{E}$. We consider the process of generating $\mathcal{S}$ as a sequential decision-making task. Our method uses a generative model and an improvement model to alternate decision-making to produce a solution. At the begining of the round, we initialize a partial solution $\mathcal{S}_0$ that $\mathcal{E} = \emptyset$. Then we use the generation model to assign a machine to an unassigned operation. At this point, the edge set $\mathcal{E}$ of $\mathcal{S}_0$ add new corresponding edge,which form the next partial solution $\hat{\mathcal{S}}_1$. Afterwards, the improvement model

will adjust the position of certain operation nodes(equivalent to changing the connections of edges in the edge set $\mathcal{E}$ in $\hat{\mathcal{S}_1}$) according to the policy to obtain a higher quality solution $\mathcal{S}_1$. The generation model and improvement model will continue to cycle through this process until all operations have been assigned. Fig 2 demonstrates the interaction process between the two models and the solution at a time step. In the following sections, we will further elaborate on the model structure and decision-making process of these two models.

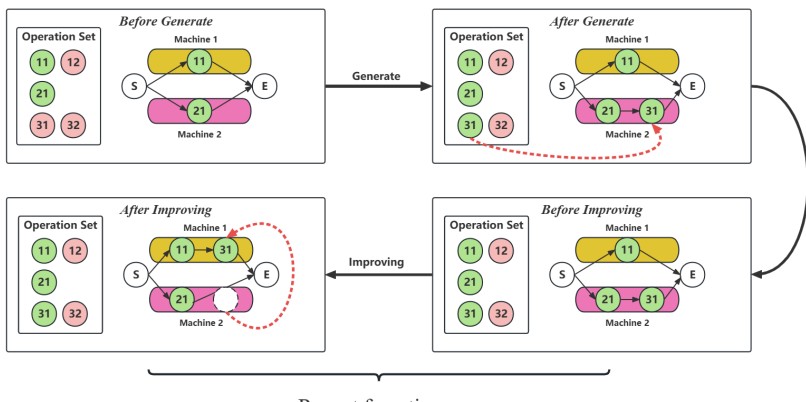

Figure 2: An overview of our framework. In a time step, the Generate Model assigns an unprocessed operation to a machine for processing and updates the solution, after which the Improve Model modifies the solution $r$ times by reinserting the processed operation.

### 3.1 MDP FORMULATION FOR FJSP

**State**  For both generate and improve step, the observation is a Graph of FJSP instance. The GAT module gets Node feature vectors from Operation Node's raw feature (Definition in section3.2) and adjacency matrix. The State vector equals avenger pooling value for all Node feature vectors which is difined as Eq.5.

**Action**  For generate step, the action's target is choosing which operation-machine pair as the next schedule. For each step, the action space size is equal to $N_j \times N_m$ ($N_j$ means number of jobs, $N_m$ means number of machine) because of the job order constraints. Different with generate, improve action will select an operation and insert it into the new position. More details on defining action vectors are presented in section 3.2.

**Transition**  A solution graph is maintained in a whole of schedule. Solution Graph consists of two parts: operation node raw feature and adjacency matrix. For generate step, when an action has been executed, a new edge between two different operation nodes will be added to the solution graph. However, for the improve step, we first need to remove the edge connected to the selected operation, and then add the corresponding edge at the insertion position.

**Reward**  For generate step, The reward function is simply equal to the difference in $C_{max}$ when the solution is updated. However, for the improve step. The reward for each time step consists of two parts: Step reward which can be calculated immediately at each time step which is as same as generate step, and global reward which can only be calculated uniformly after the enrie $r$ times improve step is finshed. It is equal to the $C_{max}$ difference between the solution before the first improve step and the solution after the last improve step divided by the number of improve steps $r$.

### 3.2 MODEL STRUCTURE

**Operation Orignal Feature Vector:**  Taking an operation node $O_{ij}$ as an example, its original features consist of seven elements. These include the assignment flag value $f_{ij}$, average processing time $t_{ij}^a$, processing start time $t_{ij}^s$, processing time $t_{ij}^p$, processing end time $t_{ij}^e$, percentile in its job

$o_{ij}^J$, and percentile in its machine processing queue $o_{ij}^M$. More specifically, the assignment flag value $f_{ij}$ equal to 0 when the operation node is not assigned to any machine for processing, otherwise it equal to 1. The average processing time is equal to the average processing time of all machines that can process the operation. The processing start time is equal to the maximum processing end time of all predecessor nodes of the operation in graph $\mathcal{S}$. The processing time is equal to the average processing time if the operation is not assigned, otherwise it is equal to the processing time required by the machine that processes the node. The processing end time is equal to the sum of the processing start time and the processing time. The job sequence is equal to the rank of the operation in its job divided by the number of operations in the job. The machine sequence is equal to the rank of the operation in its machine processing queue divided by the number of operations in the processing queue. hence,The original feature vector $u_{ij}$ of operation node $O_{ij}$ can be expressed as:

$$u_{ij} = [f_{ij}, t_{ij}^a, t_{ij}^s, t_{ij}^p, t_{ij}^e, o_{ij}^J, o_{ij}^M]$$

**GAT Module (GM):** GAT [26] is a network structure for processing graph feature extraction, which generates the features of a node by aggregating all its neighboring nodes. A GAT network layer takes the original feature vectors of the nodes and the adjacency matrix of a graph as input and outputs an aggregated feature vector for each node. In $\mathcal{S} = (\mathcal{O}, \mathcal{E}, \mathcal{C})$,we use $\mathcal{N}_{ij}^J$ to represent the neighbor nodes of $O_{ij}$ on the edge set $\mathcal{C}$, $\mathcal{N}_{ij}^M$ to represent the neighbor nodes on $\mathcal{E}$, and $u_{ij}$ to represent the feature vector of $O_{ij}$. GAT first computes the attention coefficient between $u_{ij}$ and each node $u_k$in $\{\mathcal{N}_{ij}^J \cup \mathcal{N}_{ij}^M\}$ (including $u_{ij}$ itself) as:

$$e_{ijk} = LeakyReLU(a^T[Wu_{ij}||Wu_k]) \tag{1}$$

Then the coefficients are normalized across the neighborhood using sofmax function:

$$\alpha_{ijk} = softmax(e_{ijk})$$

Finally,GAT aggregates features and apply a avitation function to get output:

$$u_{ij}^{'} = \sigma(\sum\nolimits_{k \in \{\mathcal{N}_{ij}^J \cup \mathcal{N}_{ij}^M\}} \alpha_{ijk}Wu_k) \tag{2}$$

It is worth noting that the output of a GAT network layer is also a feature vector. As a result, we can stack multiple GAT layers to form a $Gm$ to obtain more powerful feature extraction capabilities when the adjacency matrix is square. More specifically,the calculation process of the layer $i$ in $Gm$ can be represented as:

$$U^{(i)} = GAT^{(i)}(U^{(i-1)}, A)$$

**Operation Node Embedding:** We use a module $Gm_O$ consisting of $L_O$ layers of GAT stacking to obtain the embedding vector of the operation node. The $Gm_O$ takes the original feature matrix of all operation nodes $U$ and an adjacency matrix $A_O$ which constructed from all edges in $\mathcal{E}$ and $\mathcal{C}$ as input.The embedding vector of operation node can be calculated as($d$ is a hyperparameter representing the embedding dimension which the same as following):

$$E_O = Gm_O(U, A_O), E_O \in \mathbb{R}^{|\mathcal{O}| \times d} \tag{3}$$

**Insert Position Embedding:** We define the insertion position as the blank space between two adjacent operations (including start and end nodes) in the same machine processing queue. Based on this definition, for an operation in an $n \times m$ scale FJSP instance, there will be at most $n + m$ different insertion positions, so the total number of insertion positions is equal to $n_I = n \times (n+m)$. In order to use GAT to extract the features of these insertion positions, we define an adjacency matrix $A_I$ representing the insertion position with dimensions of $n_I \times n$. In the matrix, element $M_{ij} = 1$ when operation $O_j$ is adjacent to insertion position $i$, otherwise $M_{ij} = 0$.Then, we input the original feature matrix of the operation $U$ and $A_J$ into $Gm_I$ composed of a single GAT layer (since $A_J$ is not a square matrix, the GAT layers cannot be stacked), and we can output the embedding vector of all insertion position $E_I$.The embedding vector of insert position can be calculated as:

$$E_I = Gm_I(U, A_I), E_I \in \mathbb{R}^{n_I \times d}$$

**Machine Process Queue Embedding:** The embedding of the machine-processed queue is achieved through a single GAT layer due to its use of a non-square adjacency matrix for input.We

use the original features $U$ of the operation nodes and the adjacency matrix $A_M$ representing the subordinate relationship between the operation nodes and the machine as input. $A_M$ is a $|\mathcal{M}| \times |\mathcal{O}|$ ($|\mathcal{M}|$ is the number of machines, $|\mathcal{O}|$ is number of operations).For each element $M_{ij}$ of this matrix, if $O_j$ is processed on Machine $i$, then $M_{ij} = 1$, otherwise it is equal to 0.The embedding vector of machine process queue can be calculated as:

$$E_M = Gm_M(U, A_M), E_M \in \mathbb{R}^{|\mathcal{M}| \times d}$$

**Job Sequence Embedding:** Similar to the embedding of the machine processing queue, the job sequence embedding is also achieved by a single GAT layer. It takes the original feature matrix of the operation and the adjacency matrix $A_J$ representing the subordinate relationship between the operation nodes and the job as input. $A_J$ is a $|\mathcal{J}| \times |\mathcal{O}|$ ($|\mathcal{J}|$ is the number of jobs, $|\mathcal{O}|$ is number of operations).The assignment of elements is similar to that of the machine process queue.The embedding vector of job sequence node can be calculated as:

$$E_J = Gm_J(U, A_J), E_J \in \mathbb{R}^{|\mathcal{J}| \times d}$$

**Policy Model:** Our framework makes decisions based on the DuelingDQN algorithm, which estimates the advantage value of state-action pairs to make decisions. More specifically, the algorithm calculates the advantage values of all feasible actions $a \in A_t$ under state $s_t$ at step $t$ and selects the action $a_t^*$ corresponding to the maximum advantage value,as:

$$a_t^* = argmax_{a \in A_t} A(s_t, a) \tag{4}$$

For both generative model and improvement model, $s$ Obtained by average pooling of all operation node embedding vectors output by $Gm_O$,as:

$$s = meanpooling(E_O) \tag{5}$$

The action of the generated model is formed by concatenating the job embedding, the machine processing queue embedding and the operation embedding vector. More specifically, an generative action vector:

$$a_G = [E_J^i, E_O^{j(i)}, E_M^k] \tag{6}$$

represents that the first unassigned operation $O_{j(i)}$ in the job $J_i$ is inserted into the end of processing queue for machine $M_k$. Hence, at most $|\mathcal{J}| \times |\mathcal{M}|$ actions need to be considered for each generative decision. For improvement model, an action consists of operation embeddings and an insertion position embeddings.An improved action vector

$$a_I = [E_O^i, E_I^j] \tag{7}$$

means to move operation $O_i$ to insertion position $j$. Obviously, there are at most m different insertion schemes for each improvement decision. Both the generative model and the improvement model will use formula(4) to select the action to be executed in the current state $s_t$ at step $t$ on their respective feasible action sets.The advantage value function is fitted by a parametric MLP.

# 4 ALGORITHM AND TRAINING

In order to train the parameters in our framework, we propose a joint training method based on DuelingDQN algorithm [27]. We consider a complete episode as the process of the generative model and improvement model interacting with the FJSP environment. At each step, the generative model makes decision based on the previous solution and outputs action. The agent takes this action to interact with the environment and transitions to the next state to generate a new solution. Then, the improvement model outputs an action to improve it based on the solution generated by the generative model. In one step, the generative model only makes decision once because only one operation is assigned in single time step, while the improvement model makes decisions for $n_t$ times, where $n_t$ is a hand-craft function related to the order of step $t$. Therefore, in one step, the improvement model generates $n_t$ solutions and we choose the one with the minmal makespan as the state for next step. In this way, in a complete episode, for an instance with $n$ operations, the generative model generates $N_g = n$ interaction trajectories. The improvement model generates $N_i$ interaction trajectories, which $N_i = \sum_{t=0}^{n} n_t$. We store these trajectories in their respective experience pools for both models and use the DuelingDQN algorithm to train their parameters. In order to maintain the stability of the convergence of the two model parameters during joint training, we only update the parameters of one model while fixing the parameters of the other model in the same period. Then, we alternate the parameter update behavior of the two models at fixed episode intervals. We have shown the pseudocode for the training process in **Algorithm 1**.

---

**Algorithm 1** The Algorithm of GIM Training

---

Input: number of train episode $E$, The exchange interval of two model parameter training $K = 500$, The Generative Model $Model_G$ with parameters $\theta_G$, The Improvement Model $Model_I$ with parameters $\theta_I$.

Initialization: Initlize $\theta_G$, $\theta_I$, experience pool $EP_G$ and $EP_I$, Updata flag $F_u = 1$.

**for** $e = 0, 1...E$ **do**
    Sampling FJSP instance from a uniform distribution.
    Initlize: $\mathcal{S}_0$.
    **for** t=0,1... **do**
        $\hat{\mathcal{S}}_t = Model_G(\mathcal{S}_t)$
        $\mathcal{S}_t^0 = \hat{\mathcal{S}}_t$
        **for** $g = 0, 1...n_t$ **do**
            $\mathcal{S}_t^{g+1} = Model_I(\mathcal{S}_t^g)$
            Store Transition $\tau_t^g :< \mathcal{S}_t^g, \mathcal{S}_t^{g+1} >$ into $EP_I$
        **end for**
        $\mathcal{S}_{t+1} = \mathcal{S}_t^{n_t+1}$
        Store Transition $\tau_{t+1}^g :< \mathcal{S}_t, \mathcal{S}_{t+1} >$ into $EP_I$
    **end for**
    **if** $E\%K = 0$ **then**
        $F_u = -F_u$
    **end if**
    **if** $F_u > 0$ **then**
        Update the parameters $\theta_I$ for $Model_I$
    **else**
        Update the parameters $\theta_G$ for $Model_G$
    **end if**
**end for**

---

## 5 EXPERIMENT

To evaluate the performance of the proposed framework,we performed it on both FJSP instance of random synthesis and publicly available FJSP benchmarks.

### 5.1 EXPERIMENTAL SETTINGS

**Dataset:** Similar to other related work, the model proposed in this paper is trained using randomly synthesized FJSP instances. We generated 3 training instance sets of different sizes, each containing 100 randomly synthesized instances of the same size. In order to allow batch updating of neural network parameters, all instances in each instance set in the training set have the same total number of operations. In order to verify the generalization ability of the model, we also generated 7 test instance sets of different sizes. In addition, we also used (ref1),(ref1), two well-known FJSP benchmarks,which consist of several FJSP instances with different sizes. Hence, we can further verify the generalization ability of the model by its performance on these datasets.

**Configuration:** In the optimal model that we trained, the number of GAT layers $L_O = 4$ in $GAM_O$,while for other $GAMs$ which $L = 1$. The embedding dimension $d_e = 8$ for all. For all MLPs in the policy model, we set the hidden layer dimension to $d_h = 512$ and the number of hidden layers $l_h = 2$. The maximum capacity of the experience pool in the DuelingDQN network is set to 5000, and the batch size sampled from the experience pool each time is equal to 1024. The soft update parameter $\tau = 0.005$ for the Target Q networks, the discount factor $\gamma = 0.99$, and the probability of selecting a random strategy $\delta$ decays from 0.5 to 0.2 with a decay rate $\delta_d = 0.001$. (During inference, $\delta = 0$). We use the Adam optimizer to update network parameters with a learning rate of $lr = 10^{-4}$. The maximum number of training episodes $E = 200000$.The exchange interval of two model parameter training $K = 500$.

**Baselines:** For FJSP, a job sequence patching rule and a machine assignment dispatching rule are needed to complete a schedule solution. There are hundreds of dispatching rules for the job-shop scheduling problem and FJSP in literature with a wide range of performance. However, it is nearly

impossible to evaluate hundreds of dispatching rules. (ref) tested 36 rule combinations with 12 rules for job sequencing and three rules for machine selection.We selected the four best performance job sequencing rules and one machine assignment rules and combined them into four PDRs as the baseline. Four job sequencing dispatching rules include FIFO(First in First Out), MOPNR(Most Operation Number Remaining), LWKR(Least Work Remaining), MWKR(Most Work Remaining). one machine assignment dispatching rules including SPT(Shortest Processing Time). We will use these methods to compare the performance of our model with synthetic datasets and public datasets. In addition to PDRs, we also selected two meta-heuristic methods [26; 27] and two DRL-based methods [29; 30]. For the sake of data authenticity, we directly cited the best data from the articles for comparison. Therefore, we can only compare our model with these methods on public datasets.

## 5.2 COMPARISON EXPERIMENT

In the comparison experiment section, we will select the optimal model that we have trained and compare it with other methods. This optimal model was obtained through a total of 100,000 episode of training on synthetic datasets with scales of $5 \times 3$, $10 \times 5$, and $10 \times 10$.

**Performance on Random Instance**  We use the average objective value makespan, running times, and Gap to the Gurobi Solver(A commercial-grade high-performance solver tool) to evaluate the performance of various methods (including eight PDRs and our method) on synthetic test instances of different scales, summarized in **Table 1**. Through experimental data, we can clearly see that our model performs better than other PDRs on instances of different sizes, while the computational time is also in the same order of magnitude. Although the calculation time we spend will increase more as the size of the instance increases, this is not enough to make us ignore the huge advantage of the quality of the solution.

| Size | | OR-Tools | FIFO+SPT | MOPNR+SPT | LWKR+SPT | MWKR+SPT | Ours |
|------|------|------|------|------|------|------|------|
| | $C_{max}$ | **42.64** | 62.28 | 59.92 | 60.88 | 58.88 | 48.62 |
| 5×3 | Gap | 0.00% | 46.06% | 40.53% | 42.78% | 38.09% | 14.02% |
| | Time(s) | - | 0.42 | 0.39 | 0.36 | 0.28 | 0.34 |
| | $C_{max}$ | **97.42** | 159.11 | 146.15 | 153.94 | 151.08 | 108.04 |
| 10×5 | Gap | 0.00% | 63.32% | 50.02% | 58.02% | 55.08% | 10.90% |
| | Time(s) | - | 1.09 | 1.09 | 0.96 | 0.92 | 1.55 |
| | $C_{max}$ | **186.64** | 287.02 | 264.63 | 263.52 | 235 | 208.18 |
| 20×5 | Gap | 0.00% | 53.78% | 41.79% | 41.19% | 25.91% | 11.54% |
| | Time(s) | - | 3.49 | 3.43 | 3.48 | 3.37 | 8.62 |
| | $C_{max}$ | **124.26** | 239.24 | 281.34 | 220.16 | 237.84 | 145.88 |
| 10×10 | Gap | 0.00% | 92.53% | 126.41% | 77.18% | 91.41% | 17.40% |
| | Time(s) | - | 3.29 | 3.41 | 3.30 | 3.27 | 9.42 |
| | $C_{max}$ | **257.94** | 424.26 | 397.24 | 393.84 | 352.93 | 294.69 |
| 20×10 | Gap | 0.00% | 64.48% | 54.00% | 52.69% | 36.83% | 14.25% |
| | Time(s) | - | 7.235 | 7.187 | 7.170 | 7.155 | 14.77 |
| | $C_{max}$ | **288.46** | 570.49 | 729.5 | 600.8 | 696.05 | 309.40 |
| 30×10 | Gap | 0.00% | 97.77% | 152.89% | 108.28% | 141.30% | 7.26% |
| | Time(s) | - | 12.49 | 13.14 | 12.54 | 12.62 | 30.62 |
| | $C_{max}$ | - | 886.56 | 791.57 | 710.57 | 839.94 | **412.64** |
| 40×10 | Gap | - | 114.85% | 91.83% | 72.20% | 103.55% | 0.00% |
| | Time(s) | - | 18.96 | 19.63 | 18.47 | 18.90 | 59.28 |

Table 1: Performance on random instance compared to PDRs methods.

| Size | | UB* | FIFO+SPT | MOPNR+SPT | LWKR+SPT | MWKR+SPT | RGA* | 2SGA* | DRL1 | DRL2 | Ours |
|------|------|------|------|------|------|------|------|------|------|------|------|
| | $C_{max}$ | **919.50** | 974.62 | 1026.74 | 989.21 | 959.47 | 936.73 | 925.68 | 950.75 | 954.03 | 945.00 |
| v_la [33] | Gap | 0.00% | 5.99% | 11.66% | 7.58% | 4.35% | 1.87% | 0.67% | 3.40% | 3.76% | 2.77% |
| | Time(s) | - | 0.45 | 0.46 | 0.44 | 0.50 | 191.40 | 51.43 | 0.96 | - | 1.41 |
| | $C_{max}$ | - | 206.82 | 227.48 | 210.26 | 207.47 | 183 | **175.20** | 200.30 | - | 191.70 |
| mk [? ] | Gap | - | 18.05% | 29.84% | 20.01% | 18.42% | 4.45% | 0.00% | 14.33% | - | 9.42% |
| | Time(s) | - | 0.40 | 0.41 | 0.39 | 0.45 | 280.10 | 57.60 | 0.90 | - | 1.09 |

Table 2: Performance on public benchmarks. *We quoted data directly from the original literature

**Performance on Public Benhmarks**  For public datasets [31; 33], we use makespan to measure the quality of solutions obtained by different methods. In addition to PDRs, we also directly cited data

from some related works for comparison with the performance of our model. Among them, [26; 27] used meta-heuristic genetic algorithm and [29; 30] used DRL-based method. **Table 2** shows the experimental data we obtained on different public dataset. From the experimental data, it can be seen that our method still surpasses all PDRs methods in terms of solution quality. Compared to RGA and 2SGA, although our method did not outperform them in terms of solution quality, the calculation time we spent was much less than that of these meta-heuristic methods. Compared to the other two DRL-based methods, our model still has some advantages in terms of solution quality.

## 5.3 ABLATION EXPERIMENT

In order to further study the impact of the generative model and the improved model in the framework proposed in this paper on the performance of the final method, we designed some ablation experiments.

From the design of our framework, it can be seen that the generative model and the improved model can run independently, which means that we can only use the generative model to generate a complete solution or use the improved model individually to improve any feasible initial solution generated by other methods (such as random generation or PDRs). Based on this perspective, we compared the performance of different models on synthetic datasets. **Table 3** shows the specific experimental data. According to experimental data it is obviously that using either of the two models individually would result in a decrease in solution quality. Furthermore, for the improved model, different initial solutions have a significant impact on the improvement effect. The result indicates that our framework effectively improves the quality of the solution by alternating between these two models.

Another question of interest to us is whether our model can improve its generalization performance by receiving more training data. To validate this, we selected two models. One model was trained only on a synthetic instance set of size $10 \times 5$, while the other model was trained on synthetic instance sets of sizes $5 \times 3$ and $10 \times 5$. We evaluated the performance of these two models on test sets of three different sizes: $5 \times 3$, $10 \times 5$, and $10 \times 10$. The experimental results are shown in **Table 4**.

| Size | | PDRs+improve | random+improve | generate+improve | GIM |
|---|---|---|---|---|---|
| 5×3 | $C_{max}$ | 54.72 | 67.92 | 58.47 | **48.62** |
| | Gap | 12.55% | 39.70% | 20.26% | 0.00% |
| 10×5 | $C_{max}$ | 122.92 | 157.26 | 131.48 | **108.04** |
| | Gap | 13.81% | 45.56% | 21.70% | 0.00% |
| 10×10 | $C_{max}$ | 168.69 | 203.91 | 181.46 | **145.88** |
| | Gap | 15.64% | 39.78% | 24.39% | 0.00% |

Table 3: Ablation experimental results of using different methods to get a complete solution.The first three columns means using PRDs, random or generate models to generate complete solutions and then improve.

| Size | Model:10×5 | Model:10×5+5×3 |
|---|---|---|
| 5×3 | 50.95 | **48.62** |
| 10×5 | **107.21** | 108.04 |
| 10×10 | 159.86 | **145.88** |

Table 4: Model 10×5 is trained on a dataset of size 10×5 and Model 10×5+5×3 is trained on a dataset of size 10×5 and 5×3.

## 6 CONCLUSION AND DISCUSSION

In this paper, we propose an end-to-end reinforcement learning framework to solve the Flexible Job Shop Problem. We experimentally verified that our method can achieve good results. We also designed ablation experiments to further study the role of different modules in the framework. A valuable direction for future is adapting our method to other combinatorial optimization problems.

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
