# OpenReview forum: "SIMULTANEOUS GENERATION AND IMPROVEMENT: A UNIFIED RL PARADIGM FOR FJSP OPTIMIZATION"
_ICLR.cc/2024/Conference — Submitted to ICLR 2024_

### Official Review · Reviewer_kC5r · 2023-10-22

**Soundness:** 1 poor
**Presentation:** 1 poor
**Contribution:** 2 fair
**Rating:** 3
**Confidence:** 4

**Summary:**

The author proposed a deep reinforcement learning model to address the FJSP problem. This approach involves the simultaneous application of construction heuristics and improvement heuristics, enabling it to achieve better performance in shorter time on several public datasets.

**Strengths:**

- Based on the claim of paper, it seems good to use construction heuristic to construct a better partial solution and use improvement heuristic to improve the partial solution.

**Weaknesses:**

- Actions (in Section 3.1) are critical, but not defined clearly. I have no problems with actions for construction heuristics, but actions for improvement heuristics are not well defined. In Section 3.2 “Insertion Position Embedding” (P5), the definition of insertion position is undefined clearly, and why the number of choices is (n+m) for each operation. Besides, it is also unclear about why the total number of insertion positions is equal to n×(n+m). For these unclear descriptions, there is no clue to understand the proposed method. Note that in Section 3.2 “Policy Model” (P6), there is no way to understand the description “Obviously, there are at most m different insertion schemes for each improvement decision.”

- Figure 2 is confusing and unconvincing. For example, why is 31 moved to the position after 11, not before 11? If it can also be moved to that before 11, I don’t see the strategy.

- The representation of operations is inconsistent and thus makes it hard to understand how the Insert Position Embedding works (There are $O_{ij}, O_j, O_{j(i)}, O_i$ in the article).

- Lack of test results for public benchmark dataset. With comparison to [29], you should also compare with la(edata) and la(rdata). And you may test on the dataset which is referenced by [29] to improve the reliability of your method.

Presentation comments:
- Lack of spaces in many places. E.g., “Both the generative model and the improvement model will use formula(4) to select the action to be executed in the current state st at step t on their respective feasible action sets.The advantage value function is fitted by a parametric MLP”

- In section 5.1, “ In addition, we also used (ref1),(ref1),”, and “mk [? ]” in Table2. Please carefully check the content.

- In Algorithm1, “E%K” => “e%K”, the second “$EP_I$” => “$EP_G$”, etc. There should be more that you need to find out for fixing.
- There is no data in some places in the tables, such as 40x10 for OR-Tools in Table1 and mk[?] for UB* in Table2.

**Questions:**

- I am still wondering about your method for the Machine Process Queue Embedding:
Is $M_{ij} = 1$ if $O_j$ is processed on Machine $i$, or $O_j$ “can be” processed on Machine $i$? What is the concept of model designing (or why it is designed in this way)?

- It’s not clear that “Job Sequence Embedding”, if $O_{ij}$ ($j$-th operation of Job $i$) is processed then $A_J(J_i, O_{ij})$ = 1?

**Details Of Ethics Concerns:**

N.A.

---

### Official Review · Reviewer_ZW2D · 2023-10-31

**Soundness:** 2 fair
**Presentation:** 1 poor
**Contribution:** 2 fair
**Rating:** 3
**Confidence:** 3

**Summary:**

This paper proposes an RL based scheduling methods for Flexible Job Shop Problem. The approach empolys two graph neural network models, a generative model and an improvement model, which collaboratively solve the problem. At each timestep, the generative model progressively constructs a partial solution by adding a new component into the existing partial solution, and the improvement model refines this partial solution for better performance. Both models are designed to leverage inductive biases from the problem and its current partial solution, e.g., neighbor nodes from different types of edge. The models are trained end-to-end using the reward signal for each model, in an alternating manner to stabilize the learning of two models.

The proposed algorithm is evaluated with two experiments, one for synthetic datasets and the other for public benchmarks, and it showed superiority over several heuristics and DRL-based methods in terms of solution quality. Also, though the method failed to outperform the meta-heuristic algorithms, it showed comparable result while spending much less time than the meta-heuristics.

**Strengths:**

- This work proposed a new RL-based framework for solving FJSP, which combines the construction and the improvement processes so that they can be trained in end-to-end manner.
- The method utilizes different graph representation that corresponds to a single partial solution, providing each model with relevant information. This approach is both interesting and convincing.
- Ablation study for the two distinct models provides a good empirical evidence for the proposed architecture.

**Weaknesses:**

[Methods and Experiments]
- This paper doesn't provide a clear rationale or justification for the use of various embeddings. Also, there's no ablation study for these design choices.
- The method is evaluated only two public benchmarks, whereas the DRL baseline [1] has been tested on a more extensive set of benchmarks. This raises concerns about the comprehensiveness of the evaluation and potentially limits the generalizability of the proposed method's performance.
- The reported performance of DRL baseline [1] is based on the greedy selection, while the method of this paper leverages sampling for improvement steps. For fairer comparison, the results from both greedy and sampling decoding should be included. Note that sampling performance reported in [1] for v_la task is better than the proposed method, while consuming more computation time.

[Writings]
This paper has significant defects with clarity.
First of all, there are too many typos, wrong spacing and inconsistent notaions. Below are some of them:
- page 1: 'PRD' → 'PDR'
- page 2: There are many wrong spacing in Sec. 2, e.g, 'O_i,which', 'operations,O_{ij}', or "...end of production.These two ..."
- page 3: There is a wrong figure reference, '(in figure)'
- page 4: 'avenger' → 'average'
- page 4: GAT has no reference
- page 5: 'avitation' → 'activation'
- page 5: 'M_{ij}' suddenly pops up, which supposedly typo of {A_I}_{ij}, and suddenly A_J is used, which is definitely a typo.
- page 7: In the Algorithm 1, 'EP_I' → 'EP_G' for the transition of generative model.
- page 7: There are several '(ref)'s in Sec 5.1, which should have been replaced by appropriate reference.
- page 8: In the text they say they use Gurobi Solver, but they report OR-Tools in the table.
- Throughout the paper, the authors use abbreviations without declare it, e.g., DRL in page 3, GAT in page 4, and GIM in page 7 (Algorithm 1)

Moreover, the models are not clearly described, which makes it hard to fully understand the algorithm.
For example, in GAT Module section in page 5, it is unclear whether W is shared among different u's or not.
Also, the reward for each model is not stated mathematically, which introduces an ambiguity.

[1] Song, Wen, et al. "Flexible job-shop scheduling via graph neural network and deep reinforcement learning." IEEE Transactions on Industrial Informatics 19.2 (2022)

**Questions:**

- How long it takes for training?
- Why the 40 x 10 result is missing for OR-Tools?
- How can this work be extended to other scheduling or CO problems?

---

### Official Review · Reviewer_swqB · 2023-10-31

**Soundness:** 2 fair
**Presentation:** 2 fair
**Contribution:** 3 good
**Rating:** 3
**Confidence:** 3

**Summary:**

This paper proposes an end-to-end RL framework to solve the Flexible Job-Shop Problem (FJSP). The framework consists of two major components: a generation model that produces an assignment of operations that updates the partial solution, and an improving model that refines the current partial solution. By repeating the generation and improving steps until the complete solution is found, the proposed framework finds a solution for FJSP.

The authors evaluate the proposed framework with various-sized FJSP instances, and it is shown to outperform compounded Priority Dispatching Rules (PDR) but underperform Meta-heuristics (e.g., OR-tools).

**Strengths:**

- The proposed framework suggests a novel perspective for solving FJSP. Unlike the majority of iterative improving approaches that often perform improvement steps from a complete solution, the proposed framework employs "improving" actions during solution construction.

**Weaknesses:**

- The current manuscript still has room for improvement, including a more detailed explanation of the training.
- The performance evaluation of the proposed framework seems quite limited, especially as the baselines are overly simplified in Table 1.

**Questions:**

- It seems the number of improvement iterations $n_t$ would play a crucial role within the proposed framework. Could the authors provide further details on how to decide $n_t$? In the current manuscript, it is simply mentioned as a hand-crafted function depending on the iteration index $t$.
- What is GIM in Table 3? From the context, I assume it is the proposed method, but the acronym is never introduced.
- What is "Generate+improve" in Table 3? Is it different from GIM?

---

### Official Review · Reviewer_9qF7 · 2023-11-01

**Soundness:** 1 poor
**Presentation:** 1 poor
**Contribution:** 2 fair
**Rating:** 3
**Confidence:** 3

**Summary:**

The paper introduces a reinforcement learning framework tailored for the Flexible Job Shop Problem (FJSP). The methodology leverages graph neural networks, allowing the model to handle FJSP instances of varying scales. The main novelty consists of simultaneous generation and improvement: a generative model sequentially produces solutions while an improvement model refines them. Both models are trained concurrently via reinforcement learning. The approach is at least an order of magnitude faster than metaheuristics and outperforms dispatching rules and some previous RL approaches in terms of solution quality.

**Strengths:**

The tackled problem is important in several practical scheduling applications. Unlike previous approaches that either only generate solutions in one shot or only learn to improve, the proposed approach trains two models to generate and improve at the same time, which could potentially provide the “best of both worlds”, i.e., speed of one-shot generation and solution quality of improvement methods. The proposed two-stage approach and training is novel for scheduling problems to the best of my knowledge.

**Weaknesses:**

My biggest concern is that the proposed approach seems to be only applicable to a specific scheduling problem (FJSP) with no variation in terms of constraints. In the abstract, the authors state that:

> It is worth noting that this training paradigm can be readily adapted to other combinatorial optimization problems, such as the traveling salesman problem and beyond.
>

however, there is 1) no empirical evidence to justify the claim and 2) no explanation of *how* this can actually be done. For instance, how can the improvement step be applied to the traveling salesman problem (TSP), especially considering the reward function? In several combinatorial optimization problems, it is hard to define a step-wise reward function, such as in routing problems such as the TSP. Moreover, the specific design of Section 3 seems to be over-fitted to the FJSP, with new problems requiring a substantial restructuring of the model.

Another important point is that the proposed generation-improvement method is not well justified in terms of performance. There is no ablation study on just using the generator model without any improvement:

> From the design of our framework, it can be seen that the generative model and the improved model can run independently, which means that we can only use the generative model to generate a complete solution or use the improved model individually to improve any feasible initial solution generated by other methods (such as random generation or PDRs).
>

but there is no result about this; in Table 3 only the improvement method alone is shown with other models. How would the `generate` only perform? Moreover, a natural question would arise, namely why authors decided to go for a potentially more burdensome generate+improve method (in which the generator may potentially be worse due to over-reliance on the improvement model), and not just a generator. In these regards, it would be interesting to see how the same model with the generation part only would do.

The experimental section seems to be lacking some baselines - for instance, Table 1 only compares against OR-Tools and dispatching rules, but not against RGA and 2SGA and the 2 DRL baselines. Also, OR-Tools is missing the solution time, so it is difficult to assess how the proposed approach compares in solution time (given that the quality is already worse than the OR-Tools metaheuristics). Finally, no standard deviation has been reported nor multiple runs.

In terms of the quality of the paper, there is room for improvement. Aside from several typos, the writing feels sloppy, and there are missing references (`(ref)`  in the paper, mk[?] in Table 2 and more), so I would suggest some revision. More importantly, in Algorithm 1:

> Store Transition $\tau_{t+1}^g :< \mathcal{S}_t; ~ \mathcal{S}_{t+1} > \text{into} ~EP_I$
>

I believe this should be, in fact, $EP_I$ , given that $EP_G$ is not being used here.

Finally, no code has been provided to reproduce the results.

**Questions:**

1. Why did you decide to use DuelingDQN, and not actor-critic algorithms such as PPO [29] or policy gradient methods as done in MatNet [17]*?
2. As in the “weaknesses” section, how would the model perform if only the generator was trained? And what if we trained with generator+improve but only used the generator for the solution?
3. What is the number of improvement iterations $n_t$, and how was it selected?
4. How would the proposed method fare in larger-scale instances? [29] studies scale up to $100 \times 60$.

---

*Note: MatNet [17] is cited in the manuscript but not referenced throughout the text. It may be useful to at least briefly introduce the differences with the proposed method in the related works.

---

### Meta-Review · Area_Chair_5KP5 · 2023-12-02

**Metareview:**

Summary: The paper introduces a RL framework specifically designed for the Flexible Job Shop Problem (FJSP). It features an innovative approach using two models that work concurrently: one for generating solutions and another for improving them. This methodology is claimed to be significantly faster than existing metaheuristic methods and some previous reinforcement learning approaches, while also providing high-quality solutions.

Strengths: The paper's strength lies in its novel dual-model approach to solving FJSP, which combines solution generation and improvement in a concurrent training framework.

Weaknesses:

- Lack of Empirical Justification: The paper makes broad claims about adaptability to other problems but lacks empirical evidence to support these claims.  The methodology seems specifically tailored for FJSP, with no clear evidence or explanation of its adaptability to other combinatorial optimization problems.

- Missing Ablations and Baselines: There is no detailed examination of the performance of the generative model in isolation, which could provide insights into the individual contribution of each model. The paper misses certain baselines in its performance evaluation. For instance, comparisons against some existing RL-based methods and metaheuristics are not adequately presented.

- Clarity Issues: The paper has several writing and typographical errors, unclear descriptions, and missing references. Certain critical components, like the improvement step's applicability to other problems and the reasoning behind various embeddings, are not well explained.

- Incomplete Empirical Evaluations: Missing results  in comparison tables and Lack of standard deviations or multiple run results weaken the robustness of the results presented.

**Justification For Why Not Higher Score:**

All reviewers voted to reject the paper and I agree with their assessment.

**Justification For Why Not Lower Score:**

N/A

---

### Decision · Program_Chairs · 2024-01-16

Reject